# SegTime: Precise Time Series Segmentation without Sliding Window

## Abstract

Time series are common in a wide range of domains and tasks such as stock market partitioning, sleep stage labelling, and human activity recognition, where segmentation, i.e. splitting time series into segments that correspond to given categories, is often required. A common approach to segmentation is to sub-sample the time series using a sliding window with a certain length and overlapping stride, to create sub-sequences of fixed length, and then classify these sub-sequences into the given categories. This reduces time series segmentation to classification. However, this approach guarantees to find only approximate breakpoints: the precise breakpoints can appear in sub-sequences, and thus the accuracy of segmentation degrades when labels change fast. Also, it ignores possible long-term dependencies between sub-sequences. We propose a neural networks approach SegTime that finds precise breakpoints, obviates sliding windows, handles long-term dependencies, and it is insensitive to the label changing frequency. SegTime does so, thanks to its bi-pass architecture with several structures that can process information in a multi-scale fashion. We extensively evaluated the effectiveness of SegTime with very promising results.

## 1 Introduction

Time series are ordered sequences of data values and they are ubiquitous in a wide range of domains. An important tasks for time series analytics is *time series segmentation (TSS)* (Wolf et al., 2006; Gensler & Sick, 2014). It refers to the split of data into a number of non-overlapping time segments of possibly different length where each segment corresponds to a given label, that is, data in segments with the same label is similar or follows a similar pattern. Consider an example in Figure 1a where the data are generated as sinusoidal signals and can be split in several segments, where each segment has a frequency and is marked with a predefined colourful label.

TSS is essential in various impactful domains, e.g. stocks trajectories partitioning (Chakraborty et al., 2016), speaker diarisation (Wang et al., 2018), sleep stage labelling (Perslev et al., 2019), and human activity recognition (Chavarriaga et al., 2013; Roggen et al., 2010). In essence, TSS is a step-level extension of a well known time series classification problem where the latter is regarded by many as one of the most challenging problems in data mining (Yang & Wu, 2006; Esling & Agon, 2012). Indeed, TSS requires to classify segments that are not predefined in advance and there are exponentially many possible segments of a given time series.

The common approach to address TSS (Wang et al., 2018; Perslev et al., 2019; Lee et al., 2018) is to sub-sample the time series using a sliding window of a certain length and overlapping stride, to create sub-time-series of equal length, and then classify these sub-time-series into the given categories. Figure 1b exemplifies this approach by splitting the left part of the input data (Figure 1a) into 4 blue overlapping segments (windows) that slide with the given stride, and then classified each segment into dark blue, orange and brown categories. After that, these sub-time-series are concatenated together in order to determine the breakpoints between the different categories. The breakpoints are then used to segment the time series according to their categories. In this way, the problem of TSS is reduced to time series classification.

This approach is promising when the segment labelling does not change frequently, i.e., the changing frequency of segment labelling is relatively low compared to sampling rate of measuring sensors as it is in sleep staging where the segment labels are given to every 30s while the sensors sample at

100 Hz in electroencephalography (Perslev et al., 2019). However, this approach guarantees to finds only approximate breakpoints since the precise breakpoints can appear in the sub-times-series. The quality of classification with such approach highly depends on the size of the sliding window and the stride, which are difficult to tune (Yao et al., 2018). Observe in Figure 1b that the prediction differs a lot from the ground truth (the breakpoints do not match) since the windows are large, e.g., larger than the first segment of the ground truth labelling. Thus, if the segment labelling changes relatively frequently compared to the signal sampling rate, which is quite common in e.g. human activity recognition, industrial machines, driving behaviour detection, then the accuracy will drop drastically.

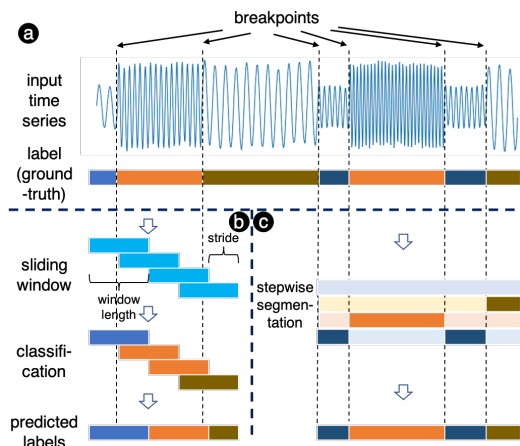

Figure 1: (a) Time series segmentation; (b) sliding window approach; (c) our approach

In this work we address the TSS problem and propose a novel neural network approach, *SegTime*, that can segment time series at time step level and excel both for fast changing labels (once per tens of samples) and slow changing labels (one per thousands of samples). This makes SegTime unique, since existing methods are typically tailored towards either fast or slow change but not for both. SegTime has many architectural highlights. We evaluated SegTime extensively against state-of-the-art baselines. To summarise, our contributions are as follows:

- *Conceptual framework.* We have several conceptual novelties: (1) we treat TSS as stepwise classification problem to achieve precise results, while in the literature TSS was commonly treated as time series classification by divide-classify-concatenate; (2) we discuss the problem of TSS for two scales of data: fast changing labels (once per tens of samples) and slow changing label (one per thousands of samples) while existing methods are typically tailored to of them only.
- *Bi-pass architecture.* We propose a bi-pass architecture with following distinguished features: (1) two core modules: our novel multi-scale skip-LSTM (long short-term memory) networks and very deep convolutional neural networks (CNN) that both are capable of capturing long-term dependencies; (2) several multi-scale pooling (depthwise separable and atrous pooling) and skipping structures in the CNN and LSTM that can process information in a multi-scale fashion, that empirically preserves the frequency-invariant representations; this enables SegTime to cope with time series of a wide spectrum of frequencies; (3) a stepwise segmentation module to replace the common practice of sliding window for TSS with step-level prediction of label sequences (Figure 1c).
- *Evaluation scheme*: (1) we propose a non-trivial transfer of a popular DeepLabv3+ net originally designed for the task of semantic segmentation to a 1D dimension for TSS, which serves as a baseline as well as part of our ablation study[1]; (2) we demonstrate benefits of our approach with three highly-optimised and sophisticated baselines on two popular datasets.

This paper is organised as follows: Sec. 2 reviews related work. Sec. 3 introduces our approach; Sec. 4 elaborates on the evaluation; Sec. 5 concludes the paper and previews future directions.

## 2 RELATED WORK

**Impact and History.** Time series segmentation is commonly used for a wide range of highly-valued domains (Bartschat et al., 2019; Waczowicz et al., 2015; ITU, 2012), and has been studied since decades. Early as 2001, Keogh et al. proposed an online algorithm for TSS applied in aeronautics and cardiography. In 2004, Keogh et al. surveyed TSS approaches for data of radio waves, exchange rate, manufacturing, civil engineering, space engineering, etc. Such studies are ubiquitous in e.g. finance industry (Yin et al., 2011; Chakraborty et al., 2016), music structure analysis (Serra et al., 2014), hydrology (Shao et al., 2010), ecology (Meschenmoser et al., 2020; Li et al., 2021), etc.

---

[1]Note this transfer of successful networks from another domain to TSS can bring significant benefit and it is non-trivial. Indeed, observe that a similar transfer of fully convolutional networks to their 1D version (U-Time) has been published in Perslev et al. (2019) and forms the core contribution of that work.

**TSS Reduced to TSC.** Surprisingly, despite of the wide range of applications and long history of study, few recent studies based representation learning (RL) directly target on precise TSS, as well as its evaluation on a step level (Gensler & Sick, 2014). Instead, TSS is often reduced to *time series classification* (TSC) problem (Wang et al., 2018; Perslev et al., 2019; Lee et al., 2018). The reason can be that precise TSS is extremely demanding, which in essence requires a step-level time series classification. Considered as one of the most challenging problems in data mining (Yang & Wu, 2006; Esling & Agon, 2012), TSC has received extensive attention in the community.

**RL-Based TSS on Data with Slow Changing Labels.** A series of RL-based works in such dividing-classification-concatenation manner are applied to the problem of sleep staging, which is to segment time series of Electroencephalogram (EEG) and electrooculogram (EOG) and label each segment with categories of sleep stage names like wake, N1, N2, rapid eye movement. The proposed method include CNN-LSTM-based networks (Supratak et al., 2017), CNN-based encoder-decoder (Perslev et al., 2019; 2021), etc. A similar problem is speaker diarisation, where the time series of speech needs to be segmented and labelled with different speaker identities. Many architectures are proposed, such as RNN-based method (Zhang et al., 2019), LSTM-based method (Wang et al., 2018), LSTM-based encoder-decoder (Fujita et al., 2019). These methods may work well on labels of low frequency change. The sampling rate of EEG/EOG signals are 100 Hz while the label is assigned to each 30min of fragment, which means the fastest label change will be slower than once per 3000 time steps. In speaker diarisation, the signal sampling rate is even higher (8kHz).

**RL-Based TSS on Data with Fast Changing Labels.** Another series of RL-based works are applied to the human activity recognition (HAR), where the label change is of higher frequency. Here the time series need to be split and human activities are assigned to each segment, such as running, walking, standing, sitting, lying, opening or closing the door, turning on or off the light (Chavarriaga et al., 2013). The signal sampling rate can be as low as 30Hz, while the label changes can be as fast as once per 20 time steps, since an action of e.g. turning off the light can happen within a short time. Methods on this topic include Classic ML, e.g. k-Nearest Neighbours, Nearest Centroid Classifier (Gjoreski et al., 2016), CNN-based (Yang et al., 2015; Gjoreski et al., 2016), CNN-LSTM-based (Wang et al., 2020; Ordóñez & Roggen, 2016), etc. Although the label change frequency is higher, they still divide the time series into fixed-length windows and perform TSC on each window.

**More Precise TSS.** A recent line of work in segmentation aims at avoiding sliding window by segmenting and classifying directly on the input time series via representation learning. Chambers & Yoder (2020) do so with a fully-connected CNN as the first layer. Another attempt (Yao et al., 2018) relies on classic ML (e.g. support vector machines) and fully-connected CNN, but the segmenting accuracy is sub-optimal and the test set here has labels of lower high frequency changes compared to the training set.

**Inspiration from Semantic Segmentation.** Studies on semantic segmentation shed light on TSS. As a dense structured prediction task, semantic segmentation predicts pixel-level labels for an image (Chen et al., 2019). This is very much in line with our goal of step-level labelling. Modules for semantic segmentation like spatial pooling pyramid (Chen et al., 2018) and ResNet (He et al., 2016) give us good inspirations. Yet, semantic segmentation is still very different, since some time series are extremely dense (e.g. 8kHz for speech). We still need to rely on time series processing modules, e.g. LSTM (Gers et al., 2000).

In summary, no prior work has investigated time series segmentation in two scales: labels of high frequency and low frequency change. Few works are dedicated a precise segmentation with step-level accuracy. These goals are achieved with our solution, SegTime.

## 3 OUR APPROACH: SEGTIME

We now formulate the problem of time series segmentation, and introduce our approach *SegTime*.

**Definitions and Problem Statement.** A (univariate) *time series* $\boldsymbol{X} = [x_1, ..., x_L]$ is an ordered sequence of real numbers, where $x_t$ is the value at the time step $t$, and $L$ is the total length. We use the term *sequence* interchangeably with *time series* when it is more convenient. $\boldsymbol{X}$ can have a stepwise corresponding *label sequence* $\boldsymbol{y} = [y_1, y_2, ..., y_L]$, which assigns a sequence of label values to $\boldsymbol{X}$; here $y_t$ is the label corresponding to $x_t$. Continuing with the example in Figure 1(a), it can be represented with a time series $\boldsymbol{X}$ of length, say $10^6$, where the first 100 elements $x_i$s correspond to the fragment of the sinusoidal signal which is marked with blue and

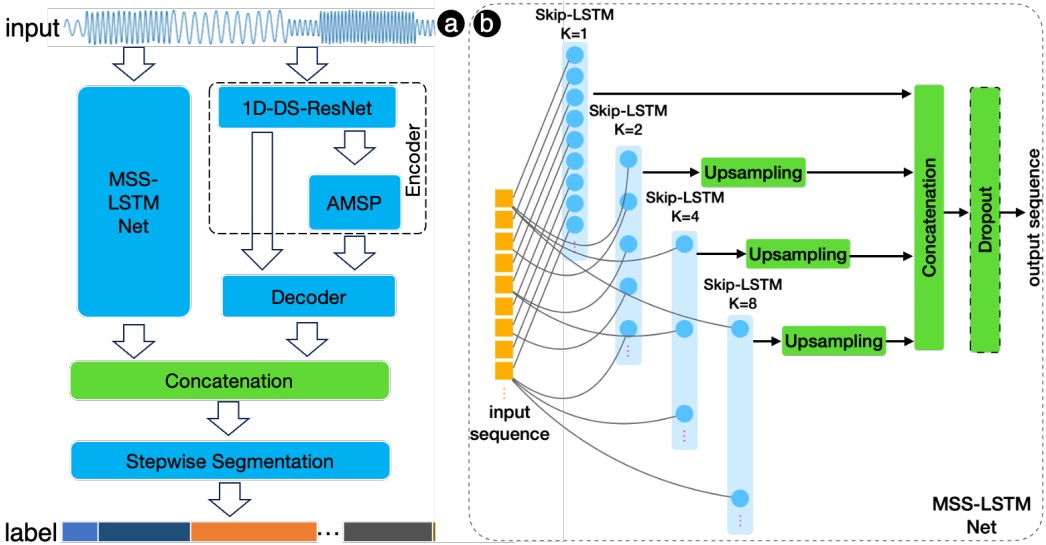

Figure 2: (a) Architecture of SegTime (An expanded overview is given in Appendix Figure 2); (b) Multi-Scale Skip LSTM Net (MSS-LSTM). Blue rounded rectangles indicate complex neutral networks, and green rounded rectangles indicate simple layers. In the skip-LSTMs, $k$ is the skipping factor. When $k = 1$, a skip-LSTM is equal to a normal LSTM.

thus all corresponding $y_i$s from 1 to 100 are equal to "blue". A *multivariate time series* of size $M$, $\boldsymbol{X}^M = [\boldsymbol{X}_1, \boldsymbol{X}_2, ..., \boldsymbol{X}_L]$, consists of $M$ univariate time series. $\boldsymbol{X}^M$ can have a stepwise corresponding label sequence $\boldsymbol{y}$, where $\boldsymbol{X}_t = [x_t^1, x_t^2, ..., x_t^M]$ is the vector of values $x_t^i$ of all $M$ time series at time step $t$, which corresponds to the label $y_t$ at time step $t$.

The goal of time series segmentation is to perform a stepwise classification of $\boldsymbol{X}$ (or $\boldsymbol{X}^M$) using a predefined set of labels $y_j$s that yields the label sequence $\boldsymbol{y}$. This in particular allows to precisely determine breakpoints, that is, the values of $j$ such that $y_j \neq y_{j+1}$.

**Architectural Overview.** SegTime consists of five modules and they are organised in a bi-pass architecture (Figure 2a). The input sequence go through two passes: an MSS-LSTM Net (Multi-scale skip LSTM) and a 1D-encoder-decoder module. The latter one consists of three sub-modules: a 1D-DS-ResNet (depth-wise separable), a AMSP (atrous multi-scale pooling) (these two form the encoder), and the decoder. The outputs of the encoder-decoder module and MSS-LSTM Net are then concatenated. Then they go through our stepwise Segmentation module, which can predict the output labels in the step-level, thus achieving precise time series segmentation.

**MSS-LSTM Net.** We propose the MSS-LSTM Net (Multi-Scale Skip LSTM, Figure 2b). In MSS-LSTM, the input sequence goes through multiple LSTM units, which have different scales of skipping (inspired by Lai et al. (2018)). We use the hyper-parameter $k$ to indicate the skipping factor (it can also be under-

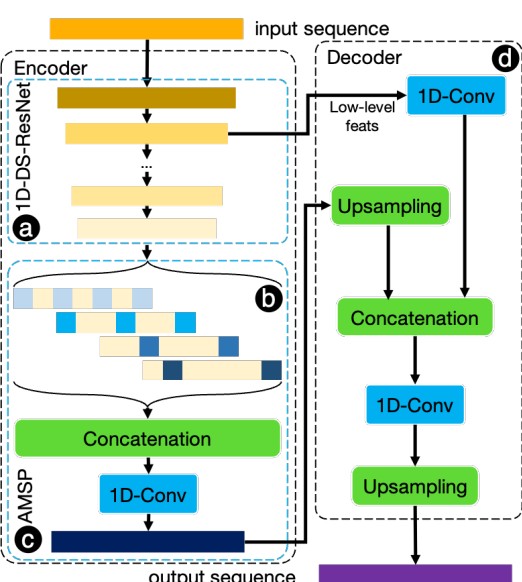

Figure 3: The 1D-encoder-decoder structure

stood as a down-sampling rate), namely by every $k$ value the Skip-LSTM takes one value. The outputs of the multiple skip-LSTM are upsampled and concatenated, and then go through an optional dropout layer. Note that the numbers of skip-LSTMs of different skipping factors are independent hyper-parameters. They can be tuned to target on data of different scales. For example, if the label changes are of high frequency, i.e. when the input sequence are sampled with relatively low

frequency, we can use more skip-LSTMs of smaller skipping factors $k$. On the other hand, if the label changes are of low frequency, i.e. when the input sequence are sampled with relatively high frequency, we can use more skip-LSTMs of larger skipping factors. One of important reasons for the excellent property of SegTime to handle time series of two scales is due to the MSS-LSTM net, since it has two important functions: (1) automatic adaption of the weights to focus on the more important frequency scale of input-output sequence pairs; (2) representation learning of temporal dependencies within the data.

**1D Encoder-Decoder.** The encoder-decoder structure (Sutskever et al., 2014) is a proven approach in various fields and has great success in the 2D problem semantic segmentation. We adopt a 1D encoder-decoder (Figure 3) with the 1D-DS-ResNet and AMSP (explained later) as the encoder and two convolutional layers as the decoder (inspired by Chen et al. (2018)). From the encoder two levels of features come out to the decoder: (1) low-level features from the early layers of 1D-DS-ResNet; (2) the output features after the AMSP. These two levels of features exploit the multi-scale information and thus contribute to the property of SegTime of handling multi-scale frequencies of data. Note that the encoder has a very deep architecture (in 1D-DS-ResNet) while the decoder has only several layers. This is because this structure can learn the representation efficiently and compresses them in a bottleneck layer.

**1D Depthwise Separable and Atrous Convolution.** A 1D depthwise separable (DS) convolution (Howard et al., 2017) separates a 1D normal convolution to two steps (Appendix Figure 1b and c): first a depthwise convolution that computes in each channel independently; the a pointwise convolution that aggregates the different channels together. Atrous convolution (a.k.a. dilated convolution, Yu & Koltun (2016)) one computes dilated convolution with skipped values (Appendix Figure 1d).

**1D-DS-ResNet.** We adopt a 1D-ResNet with depthwise separable convolutions as part of the encoder (Figure 3a). In the bi-pass architecture, we have used LSTM in the other "pass". Thus, we would like to rely on CNN here. The receptive fields in CNN are usually of limited size. Thus they often can only model local temporal patterns. To model

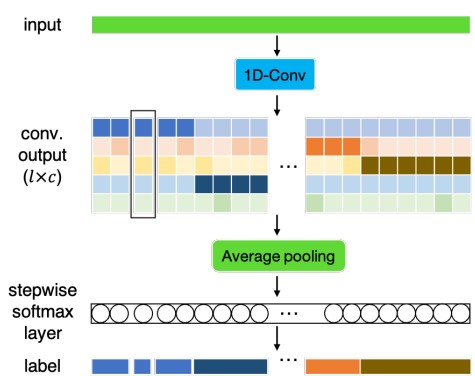

Figure 4: The stepwise segmentation. 1D-Conv: 1D convolutional layer, $l$: length of the sequence, $c$: cardinality of the labels.

long-term "memory" as LSTM does, normally more layers are required. The ResNet (He et al., 2016) is an established convolutional networks in image processing, and has been often used as a module for semantic segmentation. More importantly, ResNet can provide very deep layers of CNN, effectively mitigating the gradient vanishing problem in deep architectures.

**Atrous Multi-Scale Pooling (AMSP).** The AMSP (Figure 3c, inspired by Chen et al. (2018)) follows the same idea of handling information with multiple scales of frequency as that is in MSS-LSTM. In AMSP, a series of sequences (Figure 3b) that are down-sampled at different scales by dilation are concatenated and processed by a further 1D-convolutional layer. The output sequence serves as the high-level features, together with the low-level features, are fed into different components in the decoder to help the network automatically adapt to the unknown frequency in data.

**Stepwise Segmentation.** We adopt a 1D convolution layer followed by a an average pooling layer and a softmax layer. The number of neurons of the 1D convolution layer $c$ is equal to the cardinality of output labels, whose outputs can be seen to represent the latent labels. Note the pooling factor of the average pooling layer is dependent on the label change frequency in the output. For example, the label change in HAR can be as fast as one label for 20 time steps, and thus the pooling factor is set to 1, namely no pooling is performed. On the other hand, the label in sleep staging is assigned to every 3000 time steps, so the pooling factor can be 3000.

In summary, SegTime has several advantages: (1) it can handle time series segmentation in two scales, fast and slow changing labels, thanks to the multi-scale structures in MSS-LSTM, 1D-encoder-decoder, and AMSP; (2) SegTime reduces parameters and computation via multiple mechanisms of depthwise separable convolution, atrous convolution, and skip-LSTMs; (3) it incorporates

the long-term dependencies via MSS-LSTM and very deep architecture of 1D-DS-ResNet; (4) it allows a precise label prediction at step-level, which is due to the previous points and the stepwise segmentation module.

## 4 EVALUATION

This section first describes the datasets, then explains the experiment design, and present and discusses the results. We evaluated SegTime extensively on public datasets covering two time scales: both fast and slow changing labels. On each time scale, SegTime beats their corresponding state-of-the-art baselines, or at least achieves comparable performance.

### 4.1 DATA DESCRIPTION

An overview of the data statistics is given in Table 1. It can be seen the *Opportunity Drill* datasets have relatively fast changing labels, and the *Sleep-EDF* datasets have relatively slow changing labels (More details see Appendix Table 8 and 8).

**Opportunity Drill.** The opportunity dataset is a popular variation of Human Activity Recognition Dataset (Roggen et al., 2010; Chavarriaga et al., 2013) extensively studied for the task of HAR (Nweke et al., 2018; Yao et al., 2018; Gjoreski et al., 2016). It contains of 243 time series measured from wearable, object, and ambient sensors with a frequency of 30 Hz, including accelerometers, location sensors, switches, inertial measurement units, etc. We have chosen 113 sensors among them as a common practice in the literature (Roggen et al., 2010; Chambers & Yoder, 2020). The dataset has two groups of data, the activity of daily life (ADL) set and the drill set. The former one is recorded when the subjects are doing normal daily life activities in the morning, and the it has slower changing label (average segment length 194), while the latter one is recorded during a series of designed experiments, where the activities change intensively, thus resulting in faster label changes (average segment length 102). We select the opportunity drill set of subject 3 and 4, since many input signals of subject 1 and 2 are missing. The dataset has 7 output label sequences of body locomotion (stand, lie, sit, walk), activities (relaxing, clean up, etc.), arm gestures (cut, clean, release, etc.) and objects in the arms (bottle, salami, bread, etc.). Among which, we choose locomotion, right arm gestures, aright arm objects, and both arm gestures for evaluation, because the other label sequences almost do not change, and thus reporting on them does not bring too much. In all label features, there exists a background class (i.e. null class) where no relevant action is performed. The dataset is collected from 4 subjects, each contributed six subsets of different experiment runs. The time series lengths in these subsets range from about 25000 to 70000 time steps.

**Sleep-EDF.** This is taken from a public PhysioNet datbase (Kemp et al., 2000; Goldberger et al., 2000) often used for benchmarking sleep staging algorithms. We choose the *Sleep-EDF-39* dataset, sleep-cassette subset of 39 whole-night polysomnographic sleep recordings of healthy Causasions (taking no sleep-related medication) from age 25 to 101, because it is extensively studied in the literature (Perslev et al., 2019; 2021; Samiee et al., 2015). In sleep staging, the continuous EEG and EOG signals are segmented and the segments are categorised into stages like "Awake", "N1", "N2", "N3", and "REM". We follow the convention of sleep staging problem: we only consider the signals starting from 30min before to 30min after the first and last non-wake sleep stage; meanwhile we merge the sleep stages S3 and S4 into a single stage N3, according to the American Academy of Sleep Medicine characterisation (Conrad & AASM, 2007).

Table 1: Data statistics. Average segment length is the total number of time steps divided by the total number of segments. The smaller it is, the faster the label changes. Minimal segment length is the length of the shortest segment.

| Dataset | #Time steps | Sampling rate | Average segment length | Minimal segment length |
|---|---|---|---|---|
| Opportunity Drill | 114257 | 30Hz | 102 | 1 |
| Sleep-EDF-39 | 123723k | 100Hz | 26045 | 3000 |

### 4.2 EXPERIMENT DESIGN

We now explain our experiment design for evaluating SegTime. We conduct experiments on popular datasets with two time-scales: fast changing labels (Opportunity Drill) and slow changing labels (Sleep-EDF). We select one baseline for each of the time-scale. In addition, we adapt a state-of-the art neural network for the 2D problem of semantic segmentation. Thus, there exist four architectures to test: three baselines and SegTime. We test the four architectures on the two dataset against three performance metrics, and obtain 8 models in total.

**Baselines.** Based on the datasets of the different time scales, we select CNN Yao et al. (2018) for the Opportunity dataset (HAR task) and U-Time (Perslev et al., 2019) for the Sleep-EDF dataset (sleep staging task). In addition, we select a popular architecture for segmantic segmentation, the DeepLab (Chen et al., 2018), and adapt it to a 1D version for the time series segmentation.

*DeepConvLSTM.* This is a popular CNN-LSTM networks highly-optimised for the opportunity dataset (Ordóñez & Roggen, 2016). It has four convolutional layers and two LSTM layers. The work (Ordóñez & Roggen, 2016) adopted the common sliding window approach and tested Deep-ConvLSTM with both ADL and drill datasets. We reimplement the DeepConvLSTM and test on the drill datasets.

*U-Time.* U-Time is a feed-forward CNN-based encoder-decoder neural networks created based on U-Net (Ronneberger et al., 2015), designed to classify fixed length signals for sleep staging. U-Time has achieved state-of-the-art performance on the Sleep-EDF dataset, according to Perslev et al. (2019). It follows the classic divide-classify-concatenate strategy, and is focused on datasets with two input signals and one sequence of output labels of low frequency change (i.e. high sampling rate for input signals, 100Hz – 521Hz).

*SegTime\*.* This is a transfer and adaptation from DeepLabv3+, which is a state-of-the-art approach for semantic segmentation (Chen et al., 2018). It also adopts a CNN-based encoder-decoder architecture. In addition, it merges the depthwise separable convolution with atrous spatial pyramid pooling, and thus can exploit multi-scale information. DeepLabv3+ was not defined for time series segmentation, but we consider it very promising. Thus, we adapt its third version to a 1D architecture, adding a stepwise segmentation module with average pooling for merging outputs. We use it as one of our baselines. We name it as SegTime\* since it does not have the MSS-LSTM module.

**Performance Metrics.** We have selected three performance metrics: *accuracy*, *weighted F-score*, *weighted IoU* to evaluate the stepwise prediction of output label sequences.

*Accuracy (Acc).* Suppose the ground-truth label sequence is $\boldsymbol{y} = [y_1, ..., y_L]$, and the predicted label sequence is $\hat{\boldsymbol{y}} = [\hat{y}_1, \hat{y}_2, ..., \hat{y}_L]$. Construct the evaluation sequence $\boldsymbol{z} = [z_1, z_2, ..., z_L]$, where $z_t = 1$ if $y_t = \hat{y}_t$, otherwise $z_t = 0$. The *Acc* for each single label class is: $(TP + TN)/(P + N)$.

*Weighted F-score ($F_w$).* We use this metric as previous work (Roggen et al., 2010; Chambers & Yoder, 2020) have used it. For each label class, the F-score is calculated as $TP/(TP + \frac{1}{2}(FP + FN))$, where TP refers to true positive prediction, FP refers to false positive prediction, and FN refers to false negative prediction. The Weighted F-score $F_w$ is an average of the F-score of each label class. For the class $c$, the number of labels in the ground-truth label sequence that belong to $c$ is indicated as $N_c$. Thus the sum of all number of labels is $L$, i.e. $L = \sum_c N_c$. Weighted F-score is then calculated the weighted average of *F-score*s of all label classes as: $F_w = \frac{1}{L} \sum_c N_c Fscore_c$.

*Weighted IoU ($IoU_w$).* Weighted Mean of Intersection over Union $IoU_w$ is a common evaluation metric for semantic image segmentation (Long et al., 2015). We introduce it for time series segmentation. For each label class, the IoU is calculated as $TP/(TP + FN + FP)$, the weighted IoU is then calculated as $IoU_w = \frac{1}{L} \sum_c N_c IoU_c$.

**Cross Validation and Hyper-Parameter Selection.** We conducted 20-fold cross validation for the baselines and SegTime on the datasets. On the Sleep-EDF datasets, we follow the convention in the literature (Perslev et al., 2019): to split the data on a per-subject basis, namely to put the samples belong to the same test subject to the same train/validation/test set. We selected the hyper-parameters based on the model performance on the validation set. We set the maximum training epoch as 200, and adopt an early-stopping training strategy, when the loss on the validation set does not decrease any more for consecutive 20 epochs. The hyper-parameter selection follows a mixed scheme of limited grid search (varying one and fixing others) and human heuristics.

**Optimisation and Implementation Details.** Since the number of labels are highly imbalanced (Appendix Table 8 and 9), we adopted a on-the-fly training scheme to counter the class-imbalance issue: for each epoch, we first define the percentage $P_c$ (the chance of being sampled) for each class, and then we sample $K$ sub-sequence in total during the random sampling process, with at least $K * P_c$ sub-sequences contains the class $c$. Further more, we randomly add global and regional Gaussian noise to further increase the robustness of the methods. The source code is attached in the supplementary materials. The code is based on Python (Van Rossum & Drake, 2009) and Pytorch (Paszke et al., 2019). We ran all experiments on Amazon SageMaker with NVIDIA Tesla K80 GPU clusters.

Table 2: Results on the Opportunity Drill dataset (HAR). Loc.: locomotion, Ges.(R): gesture of the right arm, Obj.(R): object in the right arm, Ges.(B): gesture of both arms, Mean$_w$: weighted mean value of the corresponding metric.

| Methods | Metrics | Opportunity Drill with Background | | | | | Opportunity Drill without Background | | | | |
|---|---|---|---|---|---|---|---|---|---|---|---|
| | | Loc. | Ges.(R) | Obj.(R) | Ges.(B) | Mean$_w$ | Loc. | Ges.(R) | Obj.(R) | Ges.(B) | Mean$_w$ |
| Deep-Conv-LSTM | $Acc$ | 0.84 | 0.86 | 0.87 | 0.88 | 0.86 | 0.91 | 0.81 | 0.83 | 0.83 | 0.85 |
| | $F_w$ | 0.80 | 0.86 | 0.87 | 0.87 | 0.85 | 0.91 | 0.84 | 0.87 | 0.88 | 0.88 |
| | $IoU_w$ | 0.72 | 0.77 | 0.78 | 0.78 | 0.76 | 0.84 | 0.73 | 0.79 | 0.79 | 0.79 |
| U-Time | $Acc$ | 0.89 | 0.83 | 0.93 | 0.79 | 0.86 | 0.89 | 0.75 | 0.92 | 0.68 | 0.81 |
| | $F_w$ | 0.89 | 0.83 | 0.93 | 0.78 | 0.86 | 0.90 | 0.78 | 0.95 | 0.70 | 0.83 |
| | $IoU_w$ | 0.80 | 0.73 | 0.87 | 0.68 | 0.77 | 0.81 | 0.66 | 0.91 | 0.59 | 0.74 |
| SegTime* | $Acc$ | 0.90 | 0.85 | 0.90 | 0.87 | 0.88 | 0.89 | 0.77 | 0.87 | 0.84 | 0.84 |
| | $F_w$ | 0.90 | 0.85 | 0.90 | 0.87 | 0.88 | 0.90 | 0.80 | 0.91 | 0.87 | 0.87 |
| | $IoU_w$ | 0.81 | 0.74 | 0.82 | 0.77 | 0.79 | 0.81 | 0.68 | 0.84 | 0.79 | 0.78 |
| SegTime | $Acc$ | 0.89 | 0.88 | 0.88 | 0.89 | 0.88 | 0.88 | 0.81 | 0.85 | 0.86 | 0.85 |
| | $F_w$ | 0.88 | 0.87 | 0.88 | 0.89 | 0.88 | 0.89 | 0.84 | 0.89 | 0.91 | 0.88 |
| | $IoU_w$ | 0.80 | 0.78 | 0.80 | 0.81 | 0.80 | 0.80 | 0.74 | 0.82 | 0.84 | 0.80 |

We used the Adam optimiser (Kingma & Ba, 2019) with an adaptive learning rate strategy to reduce the learning rate by a factor of 10 if the validation loss stops improving for more than onsecutive 20 epochs (Note that this is easily done with ReduceLROnPlateau in Pytorch). We minimise the cross-entropy cost function with Kaiming class weights initialisation (He et al., 2015). We adopt a batch size of 32 for the opportunity dataset and 128 for the Sleep-EDF dataset.

## 4.3 RESULTS AND DISCUSSION

We present and discuss the results on datasets of two time scales, and additionally the ablation study.

**Time-Scale of Fast Changing Label.** The results of four architectures tested on the dataset with labels of high frequency change (Opportunity Drill) are shown in Table 2. The opportunity drill dataset has six output label sequences: (1) locomotion, gesture of the (2) right arm and (3) left arm, object in the (4) right arm and (5) left arm, (6) gesture of both arms. Among which, the output labels of (3) and (5) almost do not change, and thus reporting on them does not bring too much. We therefore leave them out. Important to note, the opportunity drill set is very unbalanced, and has many labels of background class (namely no activity label is assigned to the time steps). Ideally, a good architecture should not learn the dataset particularity, and be insensitive to the background class. Thus, it is also an important angle whether the network is insensitive to exclusion of background class in performance calculation.

We first compare the model when the background class is considered in the performance calculation. By looking at rows of accuracy and the column of Mean$_w$ (average weighted), it can be easily seen that SegTime outperforms SegTime* by about 1%, and SegTime significantly outperforms U-Time and CNN by 3% or more. By comparing the performance on the four output features, we see although SegTime is not always the best, it achieves consistent good performance. By looking at the output feature of Ges.(B), we see that the performance of U-Time drops drastically. We postulate the reason is that, U-Time is tailored more to slower changing labels, while the label of Ges.(B) changes very fast, and thus the performance of U-Time degrades significantly. The weighted F1-score and weighted IoU are in accordance with $Acc$, and also demonstrate that SegTime can outperform the state-of-the-art baselines for dataset with labels of fast frequency change.

We then look at the models with background class excluded from the performance calculation. We still first look and the rows of accuracy and the column of Mean$_w$, which shows all models have some deterioration in performance. Yet, SegTime still outperforms all other baselines and has relatively small deterioration, which means SegTime is more insensitive to the background class, as compared to U-Time, whose performance is worsened by about 4% when the background class is removed from calculation. If we look at the accuracy of the four output features, SegTime still has consistent good results, while the other baselines have drastic degradation on one or more output feature.

**Time-Scale of Slow Changing Label.** The results of four architectures tested on the Sleep-EDF datasets are shown in Table 3. We look at the rows of $F_w$ and the column of Mean$_w$. It can be seen that the two variants of SegTime achieve comparable results to the most optimised baseline U-Time.

Table 3: Results on the Sleep-EDF dataset for the task of sleep staging. W stands for the awake stage, REM stands for rapid eye movement stage, N1, N2 and N3 stand for the three stages of non-REM sleep, and $Mean_w$ stands for weighted mean value of the corresponding metric. The results of U-Time is directly taken from the baseline reported in Perslev et al. (2019), which has two valid significant digits.

| Dataset | Model | Metrics | W | N1 | N2 | N3 | REM | $Mean_w$ |
|---------|-------|---------|------|------|------|------|------|----------|
| Sleep-EDF-39 | Deep-Conv-LSTM | $Acc$ | 0.28 | 0.00 | 0.91 | 0.17 | 0.14 | 0.51 |
| | | $F_w$ | 0.41 | 0.00 | 0.68 | 0.21 | 0.19 | 0.44 |
| | | $IoU_w$ | 0.26 | 0.00 | 0.51 | 0.12 | 0.10 | 0.31 |
| | U-Time | $Acc$ | 0.90 | 0.46 | 0.88 | 0.83 | 0.84 | 0.83 |
| | | $F_w$ | 0.87 | 0.52 | 0.86 | 0.84 | 0.84 | 0.82 |
| | | $IoU_w$ | 0.77 | 0.35 | 0.76 | 0.73 | 0.72 | 0.71 |
| | SegTime_ | $Acc$ | 0.35 | 0.84 | 0.92 | 0.89 | 0.83 | 0.82 |
| | | $F_w$ | 0.87 | 0.40 | 0.89 | 0.82 | 0.77 | 0.82 |
| | | $IoU_w$ | 0.77 | 0.25 | 0.79 | 0.70 | 0.63 | 0.71 |
| | SegTime | $Acc$ | 0.31 | 0.89 | 0.91 | 0.86 | 0.76 | 0.82 |
| | | $F_w$ | 0.82 | 0.38 | 0.88 | 0.81 | 0.81 | 0.82 |
| | | $IoU_w$ | 0.70 | 0.24 | 0.78 | 0.68 | 0.69 | 0.70 |

Looking at the detailed accuracy on the sleep staging labels, we see that the baselines and SegTime perform differently well on the labels. Note that both U-Time and SegTime* are adapted from highly optimised and sophisticated architectures for semantic segmentation, which have achieved outstanding performance in the highly competitive domain of semantic segmentation. The baseline DeepConvLSTM is tailored to the opportunity dataset with fast changing labels. Its architecture is not deep enough to model the long-term dependencies. It is thus not surprising to see a drastic degradation of its performance on Sleep-EDF datasets. The work Perslev et al. (2019) used Mean to compare the performance, but we consider the $Mean_w$ more appropriate than the simple mean values for performance comparison, since the labels are highly unbalanced.

**Ablation Study.** We conduct ablation study to test the contribution of the two major modules of SegTime that comprise the bi-pass architecture: the MSS-LSTM and the 1D-encoder-decoder. Table 4 shows the results. The column w/o MSS-LSTM is SegTime without the MSS-LSTM net module, it is namely the SegTime*. The column w/o 1D-encoder-decoder is SegTime without the 1D-encode-decoder module. It can be seem that both modules can increase the performance of SegTime. The SegTime* already achieves very good performance, while the MSS-LSTM module further improves the performance by about 1%.

Table 4: Ablation study. Two major modules of Seg-Time (MSS-LSTM and 1D-encoder-decoder) are independently tested. w/o: without.

| Dataset | Metrics | w/o MSS-LSTM | w/o 1D-encoder-decoder | SegTime |
|---------|---------|--------------|------------------------|---------|
| Opportunity Drill | $Acc$ | 87.75% | 86.58% | 88.43% |
| | $F_w$ | 0.8763 | 0.8653 | 0.8831 |

## 5 CONCLUSION

In this work, we propose a novel neural network architecture, SegTime, that allows to do time series segmentation for datasets of a wider spectrum of time scales. We conducted experiments of comparing SegTime and highly-optimised baselines on datasets with fast changing and slow changing labels. The results show that SegTime can outperform the baselines or at least achieve comparable performance on both time scales. SegTime has multiple architectural advantages: a bi-pass architecture that combines LSTM and very deep CNN-based 1D-encoder-decoder, several multi-scale structures, depthwise separable and atrous convolution, and a stepwise segmentation module. We are very excited about our approach and its results: besides promising insights its opens exciting avenues for further research and practical industrial solutions.

## 6 REPRODUCIBILITY STATEMENT

To reproduce our results, we provide the following resources:

1. The source code, including the SegTime model details, layers, hyper-parameters, etc., link and description of the datasets in the supplementary materials.
2. Detailed model topology, hyper-parameters of the models, data descriptions, and model illustrations. In the Appendix, Table 1, 2 and 3 give a detailed view of the 1D-DS-ResNet module, Table 4 gives a detailed view of the AMSP module, and Table 5 gives a detailed view of the Decoder module, Table 6 gives a detailed view of the MSS-LSTM Net module for the SegTime evaluated on the Sleep-EDF set. Table 8 and 9 give more detailed explanation of the output features and labels for the opportunity dataset and Sleep-EDF dataset. Figure 1 schematically illustrates the 1D depthwise separable and atrous convolution. Figure 2 gives an expanded overview of the SegTime architecture.
3. Our anonymous code repository:
   `https://anonymous.4open.science/r/SegTime-0546/`.

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

# A   APPENDIX

Appendix Table 1: SegTime model topology - Module 1D-DS-ResNet of SegTime for the Sleep-EDF dataset, Part I

```
----------------------------------------------------------------
        Layer (type)           Output Shape         Param #
================================================================
           Conv1d-1         [-1, 2, 52500]              14
      BatchNorm1d-2         [-1, 2, 52500]               4
             ReLU-3         [-1, 2, 52500]               0
        MaxPool1d-4         [-1, 2, 26250]               0
           Conv1d-5         [-1, 2, 26250]               4
      BatchNorm1d-6         [-1, 2, 26250]               4
             ReLU-7         [-1, 2, 26250]               0
           Conv1d-8         [-1, 2, 26250]              12
      BatchNorm1d-9         [-1, 2, 26250]               4
            ReLU-10         [-1, 2, 26250]               0
          Conv1d-11         [-1, 4, 26250]               8
     BatchNorm1d-12         [-1, 4, 26250]               8
          Conv1d-13         [-1, 4, 26250]               8
     BatchNorm1d-14         [-1, 4, 26250]               8
            ReLU-15         [-1, 4, 26250]               0
    Bottleneck1D-16         [-1, 4, 26250]               0
          Conv1d-17         [-1, 2, 26250]               8
     BatchNorm1d-18         [-1, 2, 26250]               4
            ReLU-19         [-1, 2, 26250]               0
          Conv1d-20         [-1, 2, 26250]              12
     BatchNorm1d-21         [-1, 2, 26250]               4
            ReLU-22         [-1, 2, 26250]               0
          Conv1d-23         [-1, 4, 26250]               8
     BatchNorm1d-24         [-1, 4, 26250]               8
            ReLU-25         [-1, 4, 26250]               0
    Bottleneck1D-26         [-1, 4, 26250]               0
          Conv1d-27         [-1, 4, 26250]              16
     BatchNorm1d-28         [-1, 4, 26250]               8
            ReLU-29         [-1, 4, 26250]               0
          Conv1d-30         [-1, 4, 13125]              48
     BatchNorm1d-31         [-1, 4, 13125]               8
            ReLU-32         [-1, 4, 13125]               0
          Conv1d-33         [-1, 8, 13125]              32
     BatchNorm1d-34         [-1, 8, 13125]              16
          Conv1d-35         [-1, 8, 13125]              32
     BatchNorm1d-36         [-1, 8, 13125]              16
            ReLU-37         [-1, 8, 13125]               0
    Bottleneck1D-38         [-1, 8, 13125]               0
          Conv1d-39         [-1, 4, 13125]              32
     BatchNorm1d-40         [-1, 4, 13125]               8
            ReLU-41         [-1, 4, 13125]               0
          Conv1d-42         [-1, 4, 13125]              48
     BatchNorm1d-43         [-1, 4, 13125]               8
            ReLU-44         [-1, 4, 13125]               0
          Conv1d-45         [-1, 8, 13125]              32
     BatchNorm1d-46         [-1, 8, 13125]              16
            ReLU-47         [-1, 8, 13125]               0
    Bottleneck1D-48         [-1, 8, 13125]               0
          Conv1d-49         [-1, 8, 13125]              64
     BatchNorm1d-50         [-1, 8, 13125]              16
================================================================
```

Appendix Table 2: SegTime model topology - Module 1D-DS-ResNet of SegTime for the Sleep-EDF dataset, Part II

```
----------------------------------------------------------------
        Layer (type)            Output Shape          Param #
================================================================
            ReLU-51           [-1, 8, 13125]                0
          Conv1d-52           [-1, 8, 13125]              192
     BatchNorm1d-53           [-1, 8, 13125]               16
            ReLU-54           [-1, 8, 13125]                0
          Conv1d-55          [-1, 16, 13125]              128
     BatchNorm1d-56          [-1, 16, 13125]               32
          Conv1d-57          [-1, 16, 13125]              128
     BatchNorm1d-58          [-1, 16, 13125]               32
            ReLU-59          [-1, 16, 13125]                0
   Bottleneck1D-60          [-1, 16, 13125]                0
          Conv1d-61           [-1, 8, 13125]              128
     BatchNorm1d-62           [-1, 8, 13125]               16
            ReLU-63           [-1, 8, 13125]                0
          Conv1d-64           [-1, 8, 13125]              192
     BatchNorm1d-65           [-1, 8, 13125]               16
            ReLU-66           [-1, 8, 13125]                0
          Conv1d-67          [-1, 16, 13125]              128
     BatchNorm1d-68          [-1, 16, 13125]               32
            ReLU-69          [-1, 16, 13125]                0
   Bottleneck1D-70          [-1, 16, 13125]                0
          Conv1d-71           [-1, 8, 13125]              128
     BatchNorm1d-72           [-1, 8, 13125]               16
            ReLU-73           [-1, 8, 13125]                0
          Conv1d-74           [-1, 8, 13125]              192
     BatchNorm1d-75           [-1, 8, 13125]               16
            ReLU-76           [-1, 8, 13125]                0
          Conv1d-77          [-1, 16, 13125]              128
     BatchNorm1d-78          [-1, 16, 13125]               32
            ReLU-79          [-1, 16, 13125]                0
   Bottleneck1D-80          [-1, 16, 13125]                0
          Conv1d-81           [-1, 8, 13125]              128
     BatchNorm1d-82           [-1, 8, 13125]               16
            ReLU-83           [-1, 8, 13125]                0
          Conv1d-84           [-1, 8, 13125]              192
     BatchNorm1d-85           [-1, 8, 13125]               16
            ReLU-86           [-1, 8, 13125]                0
          Conv1d-87          [-1, 16, 13125]              128
     BatchNorm1d-88          [-1, 16, 13125]               32
            ReLU-89          [-1, 16, 13125]                0
   Bottleneck1D-90          [-1, 16, 13125]                0
          Conv1d-91          [-1, 16, 13125]              256
     BatchNorm1d-92          [-1, 16, 13125]               32
            ReLU-93          [-1, 16, 13125]                0
          Conv1d-94          [-1, 16, 13125]              768
     BatchNorm1d-95          [-1, 16, 13125]               32
            ReLU-96          [-1, 16, 13125]                0
          Conv1d-97          [-1, 32, 13125]              512
     BatchNorm1d-98          [-1, 32, 13125]               64
          Conv1d-99          [-1, 32, 13125]              512
    BatchNorm1d-100          [-1, 32, 13125]               64
================================================================
```

Appendix Table 3: SegTime model topology - Module 1D-DS-ResNet of SegTime for the Sleep-EDF dataset, Part III

```
----------------------------------------------------------------
        Layer (type)              Output Shape          Param #
================================================================
           ReLU-101            [-1, 32, 13125]                0
  Bottleneck1D-102             [-1, 32, 13125]                0
         Conv1d-103            [-1, 16, 13125]              512
    BatchNorm1d-104            [-1, 16, 13125]               32
           ReLU-105            [-1, 16, 13125]                0
         Conv1d-106            [-1, 16, 13125]              768
    BatchNorm1d-107            [-1, 16, 13125]               32
           ReLU-108            [-1, 16, 13125]                0
         Conv1d-109            [-1, 32, 13125]              512
    BatchNorm1d-110            [-1, 32, 13125]               64
           ReLU-111            [-1, 32, 13125]                0
  Bottleneck1D-112             [-1, 32, 13125]                0
         Conv1d-113            [-1, 16, 13125]              512
    BatchNorm1d-114            [-1, 16, 13125]               32
           ReLU-115            [-1, 16, 13125]                0
         Conv1d-116            [-1, 16, 13125]              768
    BatchNorm1d-117            [-1, 16, 13125]               32
           ReLU-118            [-1, 16, 13125]                0
         Conv1d-119            [-1, 32, 13125]              512
    BatchNorm1d-120            [-1, 32, 13125]               64
           ReLU-121            [-1, 32, 13125]                0
  Bottleneck1D-122             [-1, 32, 13125]                0
================================================================
Total params: 8,662
Trainable params: 8,662
Non-trainable params: 0
----------------------------------------------------------------
Input size (MB): 0.40
Forward/backward pass size (MB): 151.41
Params size (MB): 0.03
Estimated Total Size (MB): 151.84
----------------------------------------------------------------
```

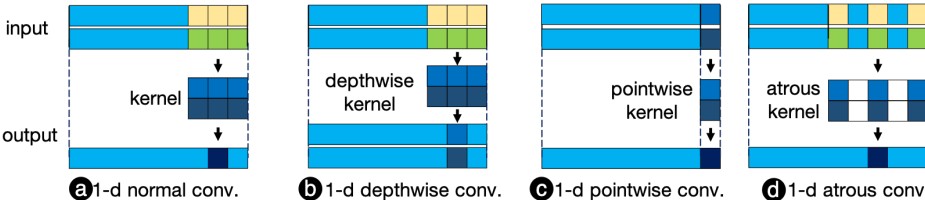

Appendix Figure 1: Depthwise separable convolution (conv.) and atrous convolution.

Appendix Table 4: SegTime model topology - Module AMSP of SegTime for the Sleep-EDF dataset

```
----------------------------------------------------------------
        Layer (type)              Output Shape          Param #
================================================================
            Conv1d-1          [-1, 8, 13125]              256
      BatchNorm1d-2           [-1, 8, 13125]               16
             ReLU-3           [-1, 8, 13125]                0
      _AMSPModule-4           [-1, 8, 13125]                0
            Conv1d-5          [-1, 8, 13125]              768
      BatchNorm1d-6           [-1, 8, 13125]               16
             ReLU-7           [-1, 8, 13125]                0
      _AMSPModule-8           [-1, 8, 13125]                0
            Conv1d-9          [-1, 8, 13125]              768
     BatchNorm1d-10           [-1, 8, 13125]               16
            ReLU-11           [-1, 8, 13125]                0
     _AMSPModule-12           [-1, 8, 13125]                0
           Conv1d-13          [-1, 8, 13125]              768
     BatchNorm1d-14           [-1, 8, 13125]               16
            ReLU-15           [-1, 8, 13125]                0
     _AMSPModule-16           [-1, 8, 13125]                0
AdaptiveAvgPool1d-17             [-1, 32, 1]                0
           Conv1d-18             [-1, 8, 1]              256
            ReLU-19              [-1, 8, 1]                0
           Conv1d-20          [-1, 8, 13125]              320
     BatchNorm1d-21           [-1, 8, 13125]               16
            ReLU-22           [-1, 8, 13125]                0
          Dropout-23          [-1, 8, 13125]                0
================================================================
Total params: 3,216
Trainable params: 3,216
Non-trainable params: 0
----------------------------------------------------------------
Input size (MB): 1.60
Forward/backward pass size (MB): 16.02
Params size (MB): 0.01
Estimated Total Size (MB): 17.64
----------------------------------------------------------------
```

Appendix Table 5: SegTime model topology - Module Decoder of SegTime for the Sleep-EDF dataset

```
----------------------------------------------------------------------
        Layer (type)               Output Shape           Param #
======================================================================
            Conv1d-1             [-1, 1, 26250]                 4
      BatchNorm1d-2             [-1, 1, 26250]                 2
             ReLU-3             [-1, 1, 26250]                 0
            Conv1d-4             [-1, 8, 26250]               216
      BatchNorm1d-5             [-1, 8, 26250]                16
             ReLU-6             [-1, 8, 26250]                 0
          Dropout-7             [-1, 8, 26250]                 0
            Conv1d-8             [-1, 8, 26250]               192
      BatchNorm1d-9             [-1, 8, 26250]                16
            ReLU-10             [-1, 8, 26250]                 0
         Dropout-11             [-1, 8, 26250]                 0
======================================================================
Total params: 446
Trainable params: 446
Non-trainable params: 0
----------------------------------------------------------------------
Input size (MB): 42057.04
Forward/backward pass size (MB): 13.42
Params size (MB): 0.00
Estimated Total Size (MB): 42070.46
----------------------------------------------------------------------
```

Appendix Table 6: SegTime model topology - Module MSS-LSTM of SegTime for the Sleep-EDF dataset

```
----------------------------------------------------------------------
        Layer (type)               Output Shape           Param #
======================================================================
        _SkipLSTM-1              [-1, 3500, 2]                40
        _SkipLSTM-2              [-1, 2100, 2]                40
        _SkipLSTM-3              [-1, 1050, 2]                40
        _SkipLSTM-4               [-1, 525, 2]                40
          Dropout-5            [-1, 105000, 2]                 0
======================================================================
Total params: 160
Total params: 160
Non-trainable params: 0
----------------------------------------------------------------------
```

Appendix Table 7: Hyper-parameters of the SegTime for the opportunity dataset and Sleep-EDF dataset.

| Parameter | Value | Comments |
|---|---|---|
| Loss function | Cross entropy | A classic loss function for classficiation. |
| Regularisation | None | |
| Class balancing | Customised | We tried to balance different classes, but not everyone, especially for those that are under-represented in the datasets. |
| Optimiser | Adam | Kingma & Ba (2019) |
| Initial learning rate | $1e^{-3}$, $5e^{-4}$ | We adopt adaptive learning rate using ReduceLROnPlateau |
| $\beta_1$ | 0.9 | in Pytorch. $1e-3$ is the initial learning rate for the |
| $\beta_2$ | 0.999 | opportunity dataset, and $5e-4$ for the Sleep-EDF dataset. |
| $\epsilon$ | $1e^{-8}$ | |
| Prediction resolution | 1, 3000 | 1 for opportunity, 3000 for Sleep-EDF. This is determined by the average pooling in the stepwise segmentation module. |
| Input dimension | 113, 1 | 113 for the opportunity dataset, 1 for the Sleep-EDF dataset |
| Input sequence length | 600, 35*3000 | 600 for the opportunity dataset, 35*3000 for the Sleep-EDF dataset |
| Conv. kernel size | 1, 3 | There exist various kernel size, please refer to the detailed model topology |
| Conv. kernel dilation size | 1, 12, 24, 36 | Multi-scale dilated (atrous) convolution |
| Max-pooling kernel size | 3 | |
| Average-pooling kernal size | 1, 3000 | 1 for the opportunity dataset because it has fast changing label, 3000 for the Sleep-EDF dataset because it |
| Padding | Same | |
| Up-sampling | Nearest neighbour | |
| Activations | ReLU | |
| Back normalisation | | |
| Parameters | 22m, 12k | 22 million for the opportunity dataset, 12k for the Sleep-EDF dataset |
| Normalisation | Min-max, robust | Min-max for the opportunity dataset, robust normalisation for the Sleep-EDF dataset. |
| Batch size | 128, 32 | 128 for the opportunity dataset, 32 for the Sleep-EDF dataset. |
| Early stopping criteria | Validation Accuracy | Accuracy computed overl all time steps |
| Model selection criteria | Validation Accuracy | We set the maximum training epoch as 200, and adopt an early-stopping training strategy, when the loss on the validation dataset does not decrease any more for consecutive 20 epochs. |
| Training epochs | 200 | |
| Examples per epoch | 1800, 800 | 1800 for the opportunity dataset, 800 for the Sleep-EDF dataset |

Appendix Table 8: Details of the opportunity dataset

| Output feature name | Encoding | Description | Label names |
|---|---|---|---|
| Locomotion | Loc. | The action of movement of the subject. | Stand, walk, sit lie |
| Right arm gesture | Ges.(R) | The gesture of the right arm of the subject. | Unlock, stir, lock, close, reach, open, sip, clean, bite, cut, spread, release, move |
| Right arm object | Obj.(R) | The object held in the right hand of the subject. | Bottle, salami, bread, sugar, dishwasher, switch, milk, drawer3 (lower), spoon, knife cheese, drawer2 (middle), table, glass, cheese, chair, door1, door2, plate, drawer1 (top), fridge, cup, knife salami, lazychair |
| Both arms gesture | Ges.(B) | The gesture of the both arms of the subject. | Open door, close door, open fridge, close fridge, open dishwasher, close dishwasher, open drawer, close drawer, clean table, drink from cup, toggle switch |

Appendix Table 9: Details of the Sleep-EDF dataset. The Sleep-EDF-39 dataset is shorten as "39" and the Sleep-EDF-153 dataset is shortened as "153".

| Label name | Encoding | Description |
|---|---|---|
| Wake | W | The condition when the subject is awake or drowsy. The brain wave is at least 50% alpha waves. |
| Non-REM1 | N1 | Short and light sleep stage. The brain wave is dominated by theta waves. |
| Non-REM2 | N2 | The brain wave is theta waves and intercepted by phenomena named as sleep spindles. |
| Non-REM3 | N3 | The brain wave is theta waves with high amplitude. |
| REM | R | Rapid eye movement occurs, The brain wave is both theta waves and alpha waves. |

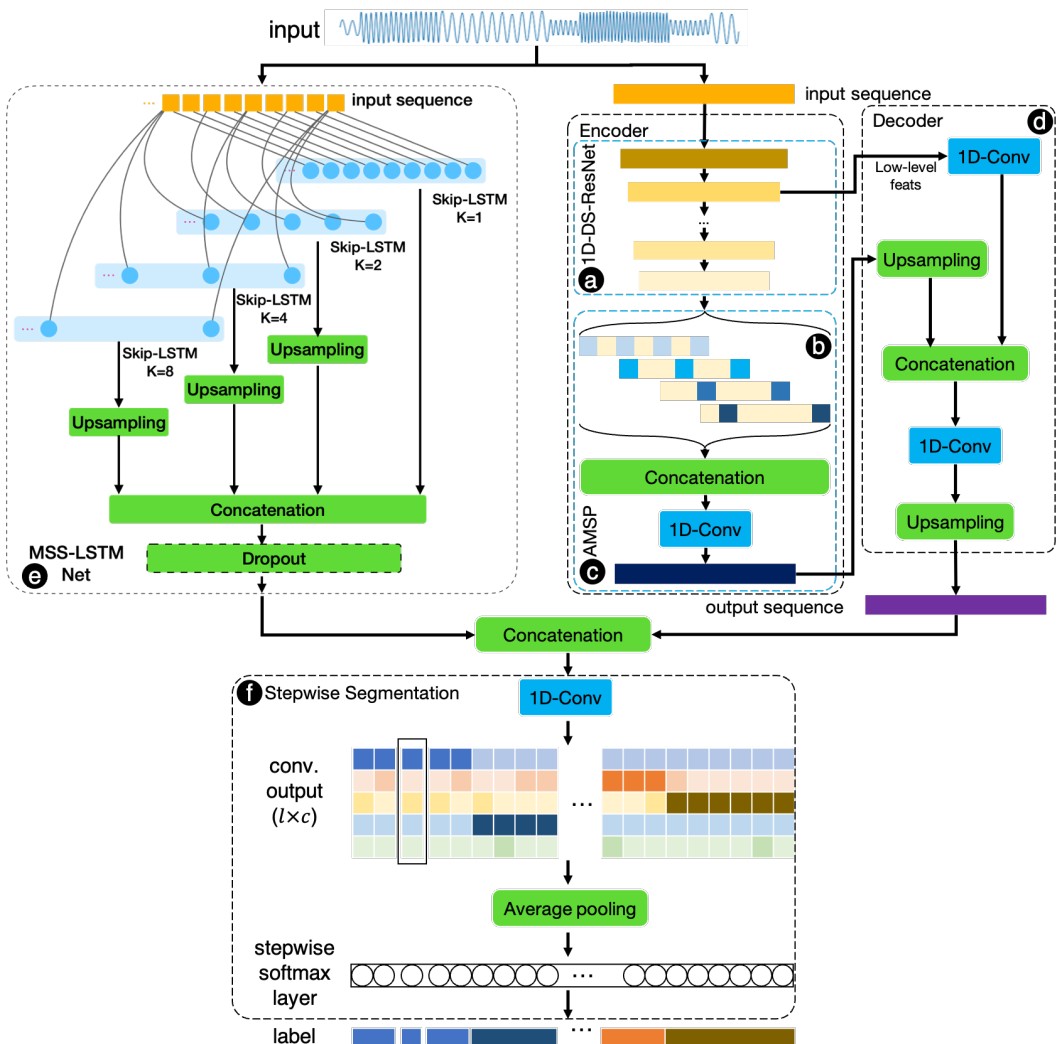

Appendix Figure 2: Expanded overview of the SegTime architecture.

