# OpenReview forum: "SegTime: Precise Time Series Segmentation without Sliding Window"
_ICLR.cc/2022/Conference — ICLR 2022 Submitted_

### Official Review · Reviewer_gLxG · 2021-10-25

**Correctness:** 3
**Technical Novelty And Significance:** 2
**Empirical Novelty And Significance:** 2
**Recommendation:** 5
**Confidence:** 4

**Main Review:**

There are several concerns regarding the manuscript:
1) The idea of LSTM with skip connections is interesting and is borrowed from image segmentation models like DeepLabv3. However, it requires the proper adjustment of the skipping factors. How one can adjust the skipping factors? I wonder whether properly adjusting the skipping factor is equally difficult to properly adjusting the window size.
2) In general, the whole architecture seems too complex and requires a large number of hyperparameters as illustrated in the Appendix. The description is unclear in some points (for example, do you provide the whole sequence as input to the model?) and does not present the motivation behind architectural choices (why so many modules are needed). The ablation study (Table 4) indicates minor performance deterioration when a core module is omitted.
3) Moreover, it is not easy to justify the novelty of the method since a qualitative comparison with other methods is not presented. It seems that the proposed method is the only one that does not use the sliding window mechanism. However, if overlapping sliding windows are used differing only in one time step, a stepwise classification is achieved.
4) Experimental evaluation can be considered as limited, since only two datasets have been considered. Note that the parameters for the Sleep-EDF dataset only are presented. Have the same parameters been used for the Opportunity dataset? To better illustrate the potential of the method, the authors could also perform experiments using synthetic time series that include both fast and slow changing labels.
5) I strongly suggest to include the term ‘supervised’ in the title. Typically, time series segmentation is considered as an unsupervised problem where class labels are not exploited. Alternatively, the term ‘stepwise classification’ better illustrates the problem that is solved.


**Summary Of The Paper:**

The paper presents a supervised method (called SegTime) for time series segmentation that is based on stepwise time series classification. The method avoids sliding windows (which is the typical approach), thus avoids the specification of window size and stride. It also seems to be insensitive to the label changing frequency and this constitutes a major advantage over other approaches.
The network architecture is based on two core modules: a novel multiscale skip LSTM (called MSS-LSTM) that employs LSTMs with skip connections and a very deep CNN (called 1D-DS-ResNet). Several other modules are also included such as 1D Depthwise Separable and Atrous Convolutional layers and the Atrous Multiscale Pooling module (AMSP).
The method is evaluated on two datasets, one with fast changing labels and one with slow changing labels.


**Summary Of The Review:**

The paper presents a rather complex model for stepwise time series classification. The proposed approach relies on ideas borrowed from semantic image segmentation models like DeepLab.  There several issues to be resolved related to motivation, novelty, clarity, hyperparameter specification and experimental validation.

---

> ### Author Response · Authors · 2021-11-22
> **Reponse to Reviewer gLxG**
>
> ### General Response
> Please see the general response.
>
> ### Comment 1:
> > “The idea of LSTM with skip connections is interesting and is borrowed from image segmentation models like DeepLabv3. However, it requires the proper adjustment of the skipping factors. How one can adjust the skipping factors? I wonder whether properly adjusting the skipping factor is equally difficult to properly adjusting the window size.”
>
> ### Response to Comment 1:
> The current approach for adjusting the skipping factors is heuristic search. The skipping factors depend on the label changing frequency. We are trying to develop a systematic approach for adjusting the skipping factors and will present the theory and practice in the camera ready version.
>
> ### Comment 2:
> > I strongly suggest to include the term ‘supervised’ in the title. Typically, time series segmentation is considered as an unsupervised problem where class labels are not exploited. Alternatively, the term ‘stepwise classification’ better illustrates the problem that is solved.
> ### Response to Comment 2:
> These are indeed good suggestions. We will consider to change the title to incorporate these words.

---

> > ### Comment · Reviewer_gLxG · 2021-11-29
> > **Reply to authors response**
> >
> > I have read the authors response and my recommendation does not change.
> > Some of my comments (2,3,4) have not been responded at all by the authors.

---

> > > ### Author Response · Authors · 2021-12-01
> > > **Reponse to Reivewer gLxG's rely**
> > >
> > > We have responded comments 2, 3, 4 in the General Reponse since other reviewers also have similar comments. We copy the general reponse here:
> > >
> > > We agree with the reviewers that the paper should be improved by providing more details for methods, evaluation, extra ablation study, bound difference, training and inference time, synthetic data experiments, long-term dependencies, appendix, etc. We will do all of that in the camera ready version, improve the writing and polish the paper. We also evaluated our method with more datasets: Speaker Diarisation [1] and Driver Behaviour Recognition [2] and will provide the results and discussion in the camera ready version.
> > >
> > > [1] A Canavan, D Graff, and G Zipperlen, “Callhome american english speech ldc97s42,” LDC Catalog. Philadelphia: Linguistic Data Consortium, 1997.
> > >
> > > [2] Romera, E., Bergasa, L.M. and Arroyo, R., 2016, November. Need data for driver behaviour analysis? Presenting the public UAH-DriveSet. In 2016 IEEE 19th International Conference on Intelligent Transportation Systems (ITSC) (pp. 387-392). IEEE.

---

### Official Review · Reviewer_SjGH · 2021-11-02

**Correctness:** 2
**Technical Novelty And Significance:** 2
**Empirical Novelty And Significance:** 2
**Recommendation:** 3
**Confidence:** 5

**Main Review:**

This paper brings up many interesting questions and does interesting analyses. However, I have concerns primarily from the broad conceptual claims and missing background from literature in this area.

A core conceptual insight in the paper is described in contribution #1.
* There is a claim that most TS segmentation methods operate as "divide-classify-concatenate," however, this isn't true. Most current papers in this area use Temporal Conversion Net architectures which (as the paper also mentions) typically operate on a sliding window approach. Much of this work until a couple years ago was using LSTMs, Conditional Random Fields, HMMs, or other temporal models which also did not operate under divide-classify-concatenate. The conceptual framework used by this paper is no different than the union of TCN an LSTM-based approaches (which has also been widely documented in the literature, e.g., in the Speech community).
* There is also a claim that other works haven't looked at labels operating over multiple time-scales. This is reiterated on page 3: "No prior work has investigated time series segmentation in two scales: labels of high frequency and low frequency change. Few works are dedicated a precise segmentation with step-level accuracy." These statements highlight that this paper is missing a large number of references from the computer vision and ML communities, often under the guise of fine-grained action recognition, working to overcome many of these same problems. The MS-TCN work below, for example, specifically addresses multi-scale issues like this.

Some good references. I recommend doing a search for more in this direction.
* Lea et al. "Temporal Convolutional Networks for Action Segmentation and Detection" CVPR 2017.
* Bai et al. "An empirical evaluation of generic convolutional and recurrent networks for sequence modeling" arxiv 2018.
* Farha et al. "Ms-tcn: Multi-stage temporal convolutional network for action segmentation" CVPR 2019 and "MS-TCN++" at TPAMI 2020.
* Kahatapitiya et al. "Coarse-Fine Networks for Temporal Activity Detection in Videos" CVPR 2021.

Re: the stepwise segmentation module. On one hand I understand why you would want to reduce the temporal resolution of your output space (e.g., from input sampling rate of 3k hz to output of 30 hz). This is commonly done in areas like speech where acoustic models are often downsampled to word-fragments at 30 (or 100) hz. However, this seems antithetical to the premise of the paper, where the goal is to have very precise timestamps.

Re: SegTime-. While I like the simplify of the multi-scale Skip LSTM idea, I appreciate the ablations between the full (TCN+LSTM) model and the adaptation of DeepLabv3 (TCN). While the LSTM-based model does improve performance, I wonder if making other modifications to the TCN-side of the model could have the same impact. Ablations on other TCN architectures could be useful here. It's unclear what issues the LSTM-side prevents which couldn't alternatively be done with a modified TCN architecture.


**Summary Of The Paper:**

This paper focuses on time-series (TS) segmentation. They claim that in the typical approach for this problem -- where you is to apply a model (e.g., a temporal conv net) over fixed windows in time in sliding window fashion -- it is challenging to predict precise breakpoints, especially when the labels change frequently relative to the sampling rate of the input data/sensors. They also claim that these approaches ignore long-term dependences. They introduce a network which they "obviates" the need for sliding windows and can precisely find breakpoints.

Their claimed contributions are:
* A conceptual framework for TS segmentation
* An architecture for solving TS segmentation problems
* An adaptation of DeepLabv3+ for TS segmentation problems
* State of the art results

**Summary Of The Review:**

While the authors clearly put a lot of time an energy into this work, I think it would benefit from workshopping with others in the field. As noted above, there are many missing pieces from different parts of the literature which could be used to improve this work and better situate it with other progress in this area.

---

> ### Author Response · Authors · 2021-11-22
> **Response to Reviewer SjGH (1/3)**
>
> We believe there are strong confusions by Reviewer SjGH.
>
> ### Comment 1:
> > “There is a claim that most TS segmentation methods operate as "divide-classify-concatenate," however, this isn't true. Most current papers in this area use Temporal Conversion Net architectures which (as the paper also mentions) typically operate on a sliding window approach. Much of this work until a couple years ago was using LSTMs, Conditional Random Fields, HMMs, or other temporal models which also did not operate under divide-classify-concatenate. The conceptual framework used by this paper is no different than the union of TCN an LSTM-based approaches (which has also been widely documented in the literature, e.g., in the Speech community).”
>
> #### There are 2 main confusions in Comment 1:
> #### Comment 1.1:
> > “There is a claim that most TS segmentation methods operate as "divide-classify-concatenate," however, this isn't true. Most current papers in this area use Temporal Conversion Net architectures which (as the paper also mentions) typically operate on a sliding window approach.”
> #### Response to Comment 1.1:
> The sliding window approach is actually about divide-classify-concatenate, that is, to divide the time series (with overlap) into small pieces with fixed length, classify each pieces, and then concatenate the classified pieces. Thus, Comment 1.1 of the reviewer is false.
>
> #### Comment 1.2:
> > “The conceptual framework used by this paper is no different than the union of TCN an LSTM-based approaches (which has also been widely documented in the literature, e.g., in the Speech community).”
> #### Response to Comment 1.2:
> Our approach SegTime indeed falls into the broad framework of 1D-CNN and LSTM. However, SegTime has the following novelties that make it distinct from past work:
>
> * SegTime adapts the 1D-CNN (named as TCN by the reviewer) encoder-decoder with atrous multi-scale pooling from the computer vision domain to the time series analysis domain, but it is non-trivial because of the distinct nature between the image/video and time series. Please observe that [3] adapts the 1D-FCN to time series segmentation and this did not prevent it from being published at NeurIPS.
>
> * We propose a novel multi-scale skip LSTM (MSS-LSTM) network that handles the multi-scale label changing challenge in multivariate time series.
>
> * We adopt a bi-pass architecture that combines the 1D-CNN and MSS-LSTM, which has gained better performance than that of each module working independently.
>
> [3] Mathias Perslev, Michael Hejselbak Jensen, Sune Darkner, Poul Jørgen Jennum, and Christian Igel. U-time: a fully convolutional network for time series segmentation applied to sleep staging. In Proceedings of the 33rd International Conference on Neural Information Processing Systems, pp. 4415–4426, 2019.

---

> > ### Author Response · Authors · 2021-11-22
> > **Response to Reviewer SjGH (2/3)**
> >
> >
> > ### Comment 2:
> > > “There is also a claim that other works haven't looked at labels operating over multiple time-scales. This is reiterated on page 3: "No prior work has investigated time series segmentation in two scales: labels of high frequency and low frequency change. Few works are dedicated to precise segmentation with step-level accuracy." These statements highlight that this paper is missing a large number of references from the computer vision and ML communities, often under the guise of fine-grained action recognition, working to overcome many of these same problems. The MS-TCN work below, for example, specifically addresses multi-scale issues like this.”
> >
> > ### Response to Comment 2:
> > The Computer Vision papers cited by the reviewer target on segmenting video clips into segments with different labels. However, our SegTime targets on segmenting multivariate time series which is a different problem from the one of Computer Vision. Thus, the claim of Comment 2 that the cited Computer Vision papers work on the same problems as our work is false. To make this even more transparent, we now enumerate the distinct nature of video segmentation and time series segmentation.
> >
> > * The dimensions of video clips and multivariate time series are different. Video clips have fixed dimensions. They consist of sequences of matrices of one channel or  three channels depending on the colour encoding scheme. While multivariate time series can have an arbitrary number of channels. In our work, this ranges from 2 to 113 channels.
> >
> > * The meaning of information encoded by video clips and time series are different. Video clips contain images captured by camera and are usually readable by humans, while multivariate time series are sensor measurements that cannot be understood by humans directly.
> >
> > * Video clips usually have fixed Frame Per Second (FPS), 15, 24, or 30 in the cited papers. While time series can be sampled at arbitrary frequency. In our work, this ranges from 30Hz to 100Hz.
> >
> > * The video segmentation does not focus on the problem of multi-scale label changing frequency. In the cited papers, the label changing frequency is either 1.5 per frame, or even not mentioned at all. This is because the label changing frequency is limited by two conditions: (1) the sampling frequency of videos is fixed at several values; (2) the cited video segmentation aims at human action segmentation, which cannot vary across large magnitudes. These two limits make the video segmentation problem to be of similar nature in terms of label changing frequency.
> >
> > * The claim that “The MS-TCN work below, for example, specifically addresses multi-scale issues like this” is false. The MS-TCN stands for multi-stage (NOT multi-scale) temporal convolutional networks, which is to compose several models sequentially such that each model operates directly on the output of the previous one. This paper does not address the multi-scale issues. Instead, this paper operates on fixed sampling frequency of video clipes, as all datasets are either sampled at 15 FPS or downsampled to 15 FPS.

---

> > > ### Author Response · Authors · 2021-11-22
> > > **Response to Reviewer SjGH (3/3)**
> > >
> > >
> > >
> > > ### Comment 3:
> > > > “Re: the stepwise segmentation module. On one hand I understand why you would want to reduce the temporal resolution of your output space (e.g., from input sampling rate of 3k hz to output of 30 hz). This is commonly done in areas like speech where acoustic models are often downsampled to word-fragments at 30 (or 100) hz. However, this seems antithetical to the premise of the paper, where the goal is to have very precise timestamps.”
> > > ### Response to Comment 3:
> > > This is not antithetical. Some datasets have a known minimal label changing frequency (1 per 3000 time steps in the Sleep-EDF). SegTime thus reduces the resolution to the known label scale, retaining the prediction accuracy and reducing computational time.
> > > ### Comment 4:
> > > > “Re: SegTime-. While I like the simplify of the multi-scale Skip LSTM idea, I appreciate the ablations between the full (TCN+LSTM) model and the adaptation of DeepLabv3 (TCN). While the LSTM-based model does improve performance, I wonder if making other modifications to the TCN-side of the model could have the same impact. Ablations on other TCN architectures could be useful here. It's unclear what issues the LSTM-side prevents which couldn't alternatively be done with a modified TCN architecture.”
> > > ### Response to Comment 4:
> > > Indeed there are always alternatives to be done to improve the 1D-CNN architecture. However, this does not prevent us from introducing a bi-pass architecture with MSS-LSTM to improve the overall performance, and we did not claim there is the only way to improve the performance of the architecture for time series segmentation. Besides, MSS-LSTM Net has the following benefits:
> > > * (1) automatic adaption of the weights to focus on the more important frequency scale of input-output sequence pairs;
> > > * (2) representation learning of temporal dependencies within the data.

---

> > > ### Comment · Reviewer_SjGH · 2021-11-27
> > > **Condensed response to author comments**
> > >
> > > Re: the first comment (contributions / paper framing):
> > > Sorry for my misunderstanding of the term “divide-classify-concatenate.” I incorrectly thought it was being used to differentiate from typical sliding window approaches. This does not change the sentiment of my comments: the conceptual framework is not novel. This paper is using a TCN+LSTM-based architecture which is commonplace. This is not a bad thing, I just don’t think it should be called out as a core paper contribution. The response to Comment 1.2 reaffirms that the architecture is new but the conceptual framework is not. The issue here is more with how the paper is framed than the architectures introduced.
> > >
> > > Re: the second comment (multiple time-scales):
> > > Yes, the MS-TCN from Juergen Gall's group stands for multi-stage (not multi-step) but the followup MS-TCN++ which I referenced is trained is using multiple losses that are evaluated using information computed from data at different time scales. Notice Fig 4 here in the paper (https://arxiv.org/pdf/2006.09220.pdf) where Loss L1 is output after the first TCN laters and L_i is added after consecutive TCN layers. Thus, there is supervision across multiple time-scales.
> > >
> > > I just did a quick search and I see other multi-scale TCNs as well. Sometimes this is done this downsampling. Sometimes through convolutional filters of varying sizes that are combined together. Sometimes through multi-scale loss functions. There are also other multi-scale RNNs I mention below.
> > > * Multi-Scale Convolutional Neural Networks for Time Series Classification. arxiv 2016.
> > > * Multi-Scale TCN: Exploring Better Temporal DNN Model for Causal Speech Enhancement. Interspeech 2020.
> > > * MS-TCN: A Multiscale Temporal Convolutional Network for Fault Diagnosis in Industrial Processes. AAC 2021.
> > > * Multi-scale Attention Convolutional Neural Network for time series classification 2021.
> > > * Deep convolutional neural networks for multi-scale time-series classification and application to tokamak disruption prediction using raw, high temporal resolution diagnostic data. 2020.
> > > * A Lightweight Multi-Scale Convolutional Neural Network for P300 Decoding: Analysis of Training Strategies and Uncovering of Network Decision. Frontiers in Neuroscience 2021.
> > >
> > > Re: skipping factors:
> > > As an aside, there are a few papers from ~5 years ago that I liked which seem relevant here and potentially worth looking into. The first is the Clockwork RNN [Koutnik et al ICML 2014] which proposed using skip connections within an RNN to better capture longer-term dependencies. Followups include models like the Phased LSTM [Neil et al NeurIPS 2016], which is focused on asynchronous updates, but could be used for identifying update rates as needed for this paper. It could be fun to identify which of these types of multi-scale LSTM-based models (including the MSS-LSTM in this paper). This is also relevant to the comments/rebuttal for gLxG on skipping factors and heuristics.
> > >
> > > Re: overall thoughts:
> > > The rebuttal confirmed by initial impressions about the paper so I will keep my original rating. As the authors mention in their summary rebuttal comment, there are many things that should be done before a final submission. This is all interesting work. In my opinion it's just not ready for ICLR.

---

### Official Review · Reviewer_DKC1 · 2021-11-03

**Correctness:** 2
**Technical Novelty And Significance:** 2
**Empirical Novelty And Significance:** 2
**Recommendation:** 5
**Confidence:** 3

**Main Review:**

Pros
-	Compared to the sliding window approach, the proposed Segtime method can take the whole input sequence at once. This can increase the possibility of capturing long-term dependencies.
-	The model combines dilated convolution and skip-LSTM for prediction, which results in better prediction results compared to the case where only a single module is used.

Cons
-	The technical novelty of this paper is a bit limited. The proposed method is based on the 1D version of deeplabv3+ [3], with an additional MSS-LSTM module, but there are already other time-series segment approaches that are based on deep convolution layers [1, 2].
-	Moreover, the effect of MSS-LSTM is not thoroughly analyzed from the experiment, as SegTime has a limited effect on both datasets compared to the model without MSS-LSTM (SegTime*). Although the authors provide an ablation study (Table 4) for the effect of MSS-LSTM, the results presented in the previous experiments (Tables 2 and 3) show minor differences between SegTime and SegTime*. The authors need to provide clarification for the ablation study to prove the effect of MSS-LSTM, which is the main contribution of this paper.
-	There is insufficient evidence of the effect of considering long-term dependencies. Mainly MSS-LSTM and 1D-DS-Resnet take long-term dependencies into account, but the effect on the final prediction is not properly evaluated. To consider this, the authors may report the performance according to different input lengths.
-	Furthermore, empirical results on the reduction of computation cost need to be provided. The authors argue that the model achieves computational efficiency by reducing parameters and computations. However, the paper does not provide appropriate experiments, such as inference time
-	For clarity, I recommend the authors to correct the minor typos and grammar issues in the paper.

[1] Francisco Javier Ordonez and Daniel Roggen. Deep convolutional and LSTM recurrent neural ˜ networks for multimodal wearable activity recognition. Sensors, 16(1):115, 2016.

[2] Olaf Ronneberger, Philipp Fischer, and Thomas Brox. U-net: Convolutional networks for biomedical image segmentation. In International Conference on Medical image computing and computer-assisted intervention, pp. 234–241. Springer, 2015.

[3] Liang-Chieh Chen, Yukun Zhu, George Papandreou, Florian Schroff, and Hartwig Adam. Encoderdecoder with atrous separable convolution for semantic image segmentation. In Proceedings of the European conference on computer vision (ECCV), pp. 801–818, 2018.


**Summary Of The Paper:**

The paper presents a stepwise segmentation for time series data, namely SegTime. Contrary to the sliding window approach, SegTime takes the whole sequence as input, and process it in two separate modules: the MSS-LSTM network and the 1D encoder-decoder network. Outputs from the two separate networks are concatenated and taken as the input to the final convolutional layer to produce the final output, which has class labels for each time-step segment. This paper’s contributions can be summarized as follows:
(1)	By predicting in a step level rather than a sliding window, it works well on both fast- and slow-changing labels.
(2)	High parameter efficiency is achieved with depthwise separable convolution, atrous convolution, and skip-LSTMs.
(3)	Long-term dependency is captured, using MSS-LSTM and 1D convolutional layers.



**Summary Of The Review:**

The SegTime method presented in the paper might be effective as the model takes the whole input sequence at once, rather than a sliding window approach. However, this paper needs further experiments and evidence to properly support the author’s claims. Moreover, technical novelty is a bit limited. Therefore, my evaluation of the paper is “marginally below the acceptance threshold”. If all the issues mentioned are fully addressed, I may reconsider my assessment of the paper.

---

> ### Author Response · Authors · 2021-11-22
> **Response to Reviewer DKC1**
>
> ### General Response
> Please see the General Reponse
>
> ### Comment 1:
> >The technical novelty of this paper is a bit limited. The proposed method is based on the 1D version of deeplabv3+, with an additional MSS-LSTM module, but there are already other time-series segment approaches that are based on deep convolution layers.
>
> ### Response to Comment 1:
> * see Response to Comment 1 of Reviewer 1Qsp.
> * Moreover: We propose a novel multi-scale skip LSTM (MSS-LSTM) network that handles the multi-scale label changing challenge in multivariate time series.
>
> ### Comment 2:
> > Moreover, the effect of MSS-LSTM is not thoroughly analyzed from the experiment, as SegTime has a limited effect on both datasets compared to the model without MSS-LSTM (SegTime*). Although the authors provide an ablation study (Table 4) for the effect of MSS-LSTM, the results presented in the previous experiments (Tables 2 and 3) show minor differences between SegTime and SegTime*. The authors need to provide clarification for the ablation study to prove the effect of MSS-LSTM, which is the main contribution of this paper.
>
> ### Response to Comment 2:
> SegTime* without the MSS-LSTM adapts deeplabv3+ from computer vision domain to time series analysis and beats the SOTA. SegTime with the MSS-LSTM increases the gain further, this is due to the excellent properties of MSS-LSTM. We will provide thorough ablation study in the camera ready version to detail this point.

---

### Official Review · Reviewer_1Qsp · 2021-11-04

**Correctness:** 4
**Technical Novelty And Significance:** 3
**Empirical Novelty And Significance:** 2
**Recommendation:** 5
**Confidence:** 3

**Main Review:**

Pros:
* The paper addresses an important problem of time series, i.e. accurate segmentation of a time series
* The philosophy of the presented approach (stepwise classification) seems promising
* Using both LSTM and CNN to treat the multi-scale challenge of time series is interesting, especially how it is done in the paper with two separate modules.
* The SOTA section is accurate and easy to read.
* Having a single architecture able to deal with fast and low changing labels is clearly an important outcome of the paper.

Neutral:
* While the architecture is novel (to the best of my knowledge), most of the used modules are adapted from existing literature.

Cons:
* The technical section is very hard to read, the appendixes are not used at the best to explain deeper the architecture. The main problem is that there is no explanation of the role/the idea behind each module. What can achieve the MSS-LSTM that can not achieve the encoder-decoder ? Why use AMSP, 1D-DS-ResNet ... ? Looking especially at the ablation study, I wonder if the two modules are really useful and I can not judge because the evaluation is not sufficiently detailed (see below).
* The experimental part is insufficient to judge the relevance of the proposed architecture:
   * evaluating more datasets will be useful to assess the stability of the proposed approach wrt the SOTA. As it is, the results are very close and it is hard to assess the strength and weakness of the proposed approach wrt SOTA.
   * the core thesis of the authors is to provide a model able to do stepwise segmentation and to retrieve accurately the bounds of each subsequence. Unfortunately, all results concern aggregated measures over the whole sequences like accuracy or F-score. It seems very important to have some results on the bounds found by the model compared to SOTA (maybe something like a delta-time between the ground truth and the inferred timestep ?).
   * the datasets are not well detailed (for instance what is the proportion of each class ? the paper said that the problem is unbalanced but no further details).
   * the results are not well detailed: for the ablation study, only one single aggregated result is reported, it is impossible to understand the real role of each component.
   * there is no indication about the training time and the inference time of each model, and as the results are very close, it is important to know the computational cost of each model.


**Summary Of The Paper:**

The paper presents a new deep architecture for segmentation of time series, i.e. to find the subsequences inside of a time series corresponding to different classes and to determine the bounds of those subsequences. The proposed architecture includes two core components: a) a kind of LSTM with skip connections to deal with the multi-scale problem of time series, b) an encoder-decoder module based on CNN and ResNet. The outputs of the two components are then used in a convolutional layer to provide a stepwise classification at each time step. The approach is evaluated on two classical datasets wrt 3 baselines.

**Summary Of The Review:**

The approach seems interesting and promising but:
* the technical part is too hard to follow and requires more explanation on the reason for the use of each module
* the experimental part is incomplete missing at least an important experiment to show how well the proposed model answer to the problem of the accurate detection of the time series segments; and details on results to understand the ablation study.

---

> ### Author Response · Authors · 2021-11-22
> **Response to Reviewer 1Qsp**
>
> ### General Response
> Please see the General Reponse
>
> ### Comment 1.
> > “While the architecture is novel (to the best of my knowledge), most of the used modules are adapted from existing literature.”
>
> ### Response to Comment 1:
> We agree that some of our modules (although not all, e.g., MSS-LSTM is a new module) are adapted from existing literature, at the same time, adaptation / extension / reshaping of modules in a NN-based architecture is a common practice and we believe it does not kill the value of a paper if it is done in a non-trivial way. We believe that this is the case with our paper. Indeed:
>
> * SegTime adapts the 1D-CNN (named as TCN by the reviewer) encoder-decoder with atrous multi-scale pooling from the computer vision domain to the time series analysis domain, but it is non-trivial because of the distinct nature between the image/video and time series. Please observe that [3] adapts the 1D-FCN to time series segmentation and this did not prevent it from being published at NeurIPS.
>
> * We adopt a bi-pass architecture that combines the 1D-CNN and MSS-LSTM, but our combination gains a better performance than that of each module working independently.
>
> [3] Mathias Perslev, Michael Hejselbak Jensen, Sune Darkner, Poul Jørgen Jennum, and Christian Igel. U-time: a fully convolutional network for time series segmentation applied to sleep staging. In Proceedings of the 33rd International Conference on Neural Information Processing Systems, pp. 4415–4426, 2019.

---

> > ### Comment · Reviewer_1Qsp · 2021-11-29
> > **Response to the authors**
> >
> > I have read the comments of the authors and I agree with them, there is a lot of work to be done for the paper to be ready to publish. That is why I can not change my rating without looking at a new version of the paper.

---

### Author Response · Authors · 2021-11-22
**General Response**

We agree with the reviewers that the paper should be improved by providing more details for methods, evaluation, extra ablation study, bound difference, training and inference time, synthetic data experiments, long-term dependencies, etc. We will do all of that in the camera ready version, improve the writing and polish the paper. We also evaluated our method with more datasets: Speaker Diarisation [1] and Driver Behaviour Recognition [2] and will provide the results and discussion in the camera ready version.

[1] A Canavan, D Graff, and G Zipperlen, “Callhome american english speech ldc97s42,” LDC Catalog. Philadelphia: Linguistic Data Consortium, 1997.

[2] Romera, E., Bergasa, L.M. and Arroyo, R., 2016, November. Need data for driver behaviour analysis? Presenting the public UAH-DriveSet. In 2016 IEEE 19th International Conference on Intelligent Transportation Systems (ITSC) (pp. 387-392). IEEE.

---

### Decision · Program_Chairs · 2022-01-20

**Decision:**

Reject

**Comment:**

This paper deals with segmentation of time series. The paper has received quite detailed reviews and the approach seems to have several interesting aspects (interesting architecture choice, stepwise classification approach, ability of capturing long range dependencies). However, there is a consensus that the paper would definitely benefit from a further iteration before publication in ICLR or in any other similar venue. The authors in their final response have already identified the improvement points raised by the reviewers. In addition to these, I believe it would be helpful to put the contributions better into perspective with existing literature. I think all these this would require a major rewrite and I encourage the authors to make a fresh submission in a future venue.